# Effects of Triheptanoin on Mitochondrial Respiration and Glycolysis in Cultured Fibroblasts from Neutral Lipid Storage Disease Type M (NLSD-M) Patients

**DOI:** 10.3390/biom13030452

**Published:** 2023-03-01

**Authors:** Nelida Inés Noguera, Daniela Tavian, Corrado Angelini, Francesca Cortese, Massimiliano Filosto, Matteo Garibaldi, Sara Missaglia, Ariela Smigliani, Alessandra Zaza, Elena Maria Pennisi

**Affiliations:** 1Department of Biomedicine and Prevention, TorVergata University of Rome, 00133 Rome, Italy; 2Unit of Neuro-Oncoematologia, Santa Lucia Foundation, IRCCS, 00143 Rome, Italy; 3Laboratory of Cellular Biochemistry and Molecular Biology, CRIBENS, Università Cattolica del Sacro Cuore, 20123 Milan, Italy; 4Neuromuscular Lab, Department of Neurosciences, University of Padova, 35122 Padova, Italy; 5Neuromuscular and Rare Neurological Disease Center, Neurology Unit, San Filippo Neri Hospital, ASL Roma 1, 00135 Rome, Italy; 6Department of Clinical and Experimental Sciences, NeMO-Brescia Clinical Center for Neuromuscular Diseases, University of Brescia, 25121 Brescia, Italy; 7Neuromuscular and Rare Neurological Disease Centre, Department of Neuroscience, Mental Health, and Sensory Organs (NESMOS), SAPIENZA University of Rome, Sant’Andrea Hospital, 00185 Rome, Italy

**Keywords:** triheptanoin, neutral lipid storage disease type M, NLSD, ATGL, lipid storage myopathy, mitochondrial respiration, glycolysis, anaplerosis, cultured fibroblast

## Abstract

Neutral lipid storage disease type M (NLSD-M) is an ultra-rare, autosomal recessive disorder that causes severe skeletal and cardiac muscle damage and lipid accumulation in all body tissues. In this hereditary pathology, the defective action of the adipose triglyceride lipase (ATGL) enzyme induces the enlargement of cytoplasmic lipid droplets and reduction in the detachment of mono- (MG) and diglycerides (DG). Although the pathogenesis of muscle fiber necrosis is unknown, some studies have shown alterations in cellular energy production, probably because MG and DG, the substrates of Krebs cycle, are less available. No tests have been tried with medium-chain fatty acid molecules to evaluate the anaplerotic effect in NLSD cells. In this study, we evaluated the in vitro effect of triheptanoin (Dojolvi^®^), a highly purified chemical triglyceride with seven carbon atoms, in fibroblasts obtained from five NLSD-M patients. Glycolytic and mitochondrial functions were determined by Seahorse XF Agylent Technology, and cellular viability and triglyceride content were measured through colorimetric assays. After the addition of triheptanoin, we observed an increase in glycolysis and mitochondrial respiration in all patients compared with healthy controls. These preliminary results show that triheptanoin is able to induce an anaplerotic effect in NLSD-M fibroblasts, paving the way towards new therapeutic strategies.

## 1. Introduction

Neutral lipid storage disease type M (NLSD-M) is an ultra-rare, recessive disorder that causes severe skeletal and cardiac muscle damage with lipid accumulation in virtually all tissues of the body, because of the mutations in the *PNPLA2* gene-coding adipose triglyceride lipase (ATGL). To date, no therapy is available for this very disabling disease, and the pathogenesis is still largely unknown. A decrease in the lipolytic action of the ATGL enzyme induces the enlargement of cytoplasmic lipid droplets (LDs) and a reduction in mono- and diglycerides for energy production in the cell. The cytoplasmic accumulation of the lipids alone does not totally explain the muscle atrophy seen in patients [1]. With this assumption, some studies have been carried out to evaluate whether altered functioning in the aerobic mitochondrial metabolism was also involved in the pathogenesis of the disease [2].

Mitochondria generate up to 90% of the energy in the cell, producing adenosine triphosphate (ATP) in the β-oxidation process, through the metabolism of fatty acids (FAs). This pathway also produces molecules that are used as cellular structural components for post-translational modifications of proteins and in signaling cascades [3]. In other muscle disorders due to defects in the lipid metabolism, such as very-long-chain acyl-CoA dehydrogenase deficiency (VLCAD), long-chain 3-hydroxyacyl-CoA dehydrogenase deficiency (LCHAD), trifunctional protein deficiency (TFP), and carnitine palmitoyl transferase II deficiency (CPT II), medium-branched chain fatty acids were successfully utilized as anaplerotic treatments. The MCT diet plus carnitine supplementation in these disorders stimulates citric acid cycle function and ATP production, enhances gluconeogenesis and urea cycle function, and maintains intra-mitochondrial homeostasis [4].

Triheptanoin (Dojolvi^®^) is a synthetic medium-chain triglyceride (MCT) developed by Ultragenyx Pharmaceutical Inc., Food-and-Drug-Administration approved, for use in the treatment of inherited metabolic disorders [5]. Triheptanoin is a triglyceride of three odd-chain fatty acids (heptanoate, C7), cleaved by intestinal lipases, and heptanoate is absorbed by the gastrointestinal tract [6]. Intracellularly, C7 crosses the mitochondrial membranes without an active transport system and can enter the β-oxidation cycle. The rationale to use triheptanoin in patients with NLSD-M is based on the hypothesis that energy deficiency in fatty acid oxidation may be exacerbated by the depletion of catalytic intermediates in the TCA and would, thus, benefit from the anaplerotic effect of triheptanoin, as well as to its ability to bypass metabolic block induced by ATGL deficiency. Vockley et al. demonstrated that triheptanoin treatment is associated with reduced major clinical events, such as rhabdomyolysis, cardiomyopathy, and hypoglycemia in patients with long-chain fatty acid oxidation disorders [7]. However, in studies on mice, using long-term supplementation of triheptanoin induces de novo biosynthesis and elongation of fatty acids in both WT and VLCAD-/- mice, altering the hepatic and cardiac fatty acid composition to no physiological profiles, characterized by a strong reduction in polyunsaturated fatty acids and a marked increase in monounsaturated species [8]. These results suggest that the time of application and the dose should be individually adapted to meet the energy demand of the patient. Our study explores the potential beneficial effect of triheptanoin on metabolism on NLSD-M cells in vitro.

## 2. Materials and Methods

### 2.1. Patients

Two women and three men affected by NLSD-M, from the Italian NLSD Group [9], aged 50 y and 68 y, underwent skin biopsy after informed consent was provided. Clinical and genetic data of these patients are briefly summarized in Table 1. Still, ambulatory patients were considered less severely affected than non-ambulatory patients, and patients with weakness in only a few muscles were considered to have very mild disease. Control fibroblasts were obtained after informed consent, from skin biopsies of 3 subjects at different ages (12, 58, and 73 years old), undergoing orthopedic surgery. Controls were tested for increases in CK. Myopathic signs and symptoms were excluded by neurological evaluation. Fibroblasts obtained from controls were tested for ATGL gene mutations. The study was conducted in accordance with the Helsinki Declaration.

### 2.2. Characterization of Metabolism via Seahorse XF Agylent Technology

Fibroblasts from the NLSD-M patients and the normal subjects were cultured in DMEM high-glucose medium (Euroclone, MI, Italy), 10% fetal bovine serum (FBS) (GIBCO-BRL), 20 mM Hepes, 100 U/mL penicillin, and 100 µg/mL streptomycin (GIBCO-BRL). Cultures were maintained at 37 °C in a 5% CO_2_ humidified incubator. Fibroblasts were treated with triheptanoin at 25 µM for one week before analysis. At the time of the experiment, 25 µM of triheptanoin or DMSO was added.

Using a Seahorse Bioscience XFe96 analyzer (Agilent Technologies, Santa Clara, CA, USA), mitochondrial and glycolytic function were assessed, and extracellular flux assay kits were used to measure oxygen consumption (OCR) and glycolytic flux (ECAR), as previously described [10,11]. Briefly, fibroblasts were seeded and incubated overnight at a density of 10 × 10^4^ cells/well using cell culture microplates, four wells for each condition (XFe Seahorse, Agilent Technology, Santa Clara, CA, USA). Before the start of the test, the cells were incubated in XF-DMEM medium, pH 7.4 (Agilent Technology, Santa Clara, CA, USA) at 37 °C without CO_2_ for 1 h. To evaluate mitochondrial function, the oxygen consumption rate (OCR) was measured using the Seahorse Bioscience XF Cell Mito Stress Test (Agilent Technology, Santa Clara, CA, USA). Mitochondrial oxidative phosphorylation (OXPHOS) was analyzed under basal conditions, in the presence of 2 µM oligomycin, 1 µM carbonyl cyanide-4 (trifluoromethoxy) phenylhydrazone (FCCP), and 0.5 mM rotenone/antimycin A (R/A). Oligomycin inhibits ATP synthase (V complex), as it decreases the flow of electrons through the electron transport change (ETC), resulting in a reduction in mitochondrial respiration. This reduction in OCR is linked to cellular ATP production. FCCP is a decoupling agent that interrupts the potential of the mitochondrial membrane. Rotenone, a complex I inhibitor, and antimycin A, a complex III inhibitor, shut down mitochondrial respiration, enabling the calculation of non-mitochondrial respiration. As a result, the oxygen consumption of complex IV reaches a maximum and can be used to calculate respiratory reserve capacity, defined as the difference between maximum respiration and baseline respiration. Respiratory reserve capacity is a measure of the cell’s ability to respond to increased energy demand or under stress.

The extracellular acidification rate (ECAR) caused by the conversion of glucose into pyruvate, and subsequently into lactate, is indicative of glycolysis. The Glycolysis Rate Assay measured glycolytic rates for basal conditions and compensatory glycolysis following mitochondrial inhibition after the injection of 0.5 mM R/A. The final injection is 2-deoxy-glucose (2-DG), a glucose analogue, which inhibits glycolysis through competitive binding with glucose hexokinase. The resulting reduction in ECAR confirms that the ECAR produced in the experiment is due to glycolysis.

### 2.3. MTT Assay

Thus, 2 × 10^4^ cells for each NLSD-M line were washed with phosphate-buffered saline (DPBS) 24 h after seeding and then fed with DMEM supplemented with 12.5 or 25 μM triheptanoin. Triheptanoin was diluted in DMSO. Cells grown in the medium supplemented only with DMSO were designated as controls. At the end of the short treatment (24 h, 48 h, and 72 h), the Thiazolyn Blue Tetrazolium Bromide (MTT, Merck, Darmstadt, Germany) assay was used to quantify cell viability and proliferation according to data sheet. Absorbance was measured with VICTOR^®^ Nivo™ Reader. (PerkinElmer, La Jolla, CA, USA). Each experimental sample was tested in triplicate.

### 2.4. Triglyceride Intracellular Content

NLSD-M and control fibroblasts were cultured in DMEM supplemented with 12.5 or 25 μM triheptanoin. The cellular triacylglycerol (TAG) accumulation was quantified before and after 9-day treatment using Triglyceride Quantification Colorimetric kit (Biovision Diagnostics, LLC, Collinsville, USA) according to the instructions. In brief, 1 × 10^6^ cells were homogenized in 1 mL solution containing 5% NP-40 in water and then incubated in a reaction mix for 1 h in dark conditions. The absorbance was measured at 570 nm with VICTOR^®^ Nivo™ Reader (PerkinElmer, La Jolla, California, USA).

### 2.5. Immunofluorescence Analysis

2 × 10^5^ NLSDM fibroblasts were seeded on coverslips in an Earle’s MEM culture medium and allowed to adhere for 12 h. Then, the culture medium was supplemented with 12.5 or 25 μM triheptanoin. After 9 days, the cells were fixed on coverslips using 3% paraformaldehyde, rinsed with distilled water, and incubated with Nile Red (NR) solution for 20 min in the dark. NR staining solution was freshly prepared in DPBS (1:100 *v*/*v*) from a saturated solution (1 mg/mL) in DMSO. Images were obtained using a Leica DM5000B microscope equipped with a 40× objective (NR staining: excitation at 450–490 nm) and analyzed using the public domain Java image processing program ‘WCIF ImageJ 1.35j’ (developed by W. Rasband, NIH, Bethesda, MD, USA). This software allows us to isolate components with the same wavelength and to measure and quantify different parameters such as area, numbers of selected units (LDs, in this case), and pixels per cell; 15 inches [2] was chosen as threshold value for LD area to discard fluorescent emissions due to impurity.

### 2.6. Statistical Analysis

Data were analyzed using GraphPad Prism 6 (GraphPad Software Inc., San Diego, CA, USA). Statistical analysis was performed using the Mann–Whitney test and Student’s *t*-test. Statistical significance was established at *p* < 0.05.

## 3. Results

### 3.1. Mitochondrial Respiration

Basal and maximal respiration, proton leak, ATP, and spare respiratory capacity were determined as attributes of oxidative phosphorylation (OXPHOS). Cellular basal respiration represents the energetic demand of the cell under baseline conditions; maximal respiration measures maximal oxygen consumption rate that cells can achieve; spare respiratory capacity measures the difference between the ATP produced by OXPHOS at basal and maximal activity. These parameters were significantly increased after treatment with triheptanoin in the five patients analyzed, indicating that triheptanoin ameliorates the capability of NLSD-M fibroblasts to respond to an energetic demand (Figure 1A and Table 2).

The improvement in mitochondrial respiration allowed for an increase in ATP production in fibroblasts from patients NLSD 1, NLSD 3, and NLSD 5. The other two patients did not show an enhancement in ATP production, probably because of a slight increase in proton leak levels (Figure 1A and Table 2). To produce energy, the electrons from substrate oxidation are passed through the respiratory chain on the inner mitochondrial membrane (IMM), and the flux is maintained by proton pumping against gradient from the inner mitochondrial matrix to the intermembrane space via respiratory complexes I, III, and IV. All three control samples did not show variations in any attributes of OXPHOS analyzed (Figure 1B and Table 2). Interestingly, basal OXPHOS mean values are normalized after triheptanoin treatment in NLSD-M patients’ fibroblasts, whereas there is no difference in normal controls’ fibroblasts. Triheptanoin treatment also improved maximal respiration, spare respiratory capacity, and ATP production. The trend was pronounced but the differences were not significant, probably because of the low number of samples analyzed (Figure 1C and Table 3). Moreover, excluding the NLSD 4 patient whose molecular defect completely abolishes ATGL enzymatic activity, the difference is significant in maximal respiration and in SRC (Figure 1D), suggesting that the treatment might enhance residual enzymatic activity.

### 3.2. Glycolytic Respiration

Glucose in cells is converted to pyruvate and then to lactate in the cytoplasm, or to CO2 and water in the mitochondria. The conversion of glucose to lactate results in a net production of protons into the extracellular medium, which is detected by the XF Analyzer as extracellular acidification rate (ECAR). The rates of glycolysis were determined as a percentage increase in ECAR after the addition of the complex I and complex III mitochondrial inhibitors rotenone/antimycin and of 2-Deoxy Glucose, an inhibitor of glycolysis to confirm pathway specificity. Compensatory glycolysis is the rate of glycolysis in cells following the addition of mitochondrial inhibitors, which effectively block oxidative phosphorylation and drive compensatory changes in the cell to use glycolysis to meet the cells’ energy demands. Consistently, we found that triheptanoin treatment promoted the activation of the aerobic glycolysis pathway in NLSD fibroblasts. Indeed, we observed a significant basal glycolysis increase in NLSD 2, 3, 4, and 5, while NLSD1 presented an increase in the compensatory glycolysis values (Figure 2A,C and Table 4). These effects were not observed in controls (Figure 2B,C and Table 4).

### 3.3. Cellular Viability and TAG Accumulation Study

To investigate the effect of triheptanoin supplementation on proliferation, a time course of 24 h, 48 h, and 72 h was performed on cells from NLSD 2, NLSD3, NLSD4, and NLSD5 patients. After 48 h, cell viability significantly increased in the NLSD 2 cell line cultured with 12.5 μM triheptanoin (*p* ≤ 0.05), in comparison with untreated cells. After 72 h, NLSD 2 and 3, treated with 12.5 or 25 μM triheptanoin, showed significantly higher viability than those of untreated cells (*p* ≤ 0.05). For NLSD 4 and 5 fibroblasts, no significant differences were observed (Figure 3A). Finally, intracellular TAG content was detected before and after 9-day treatment. The results showed that triheptanoin did not decrease neutral lipid abnormal storage. On the contrary, NLSD-M cells slightly increased the TAG amount stored in cytoplasmic lipid droplets (Figure 3B). Immunofluorescence analysis of LDs in NLSD-M fibroblasts confirmed data obtained by TAG evaluation. After treatment, the cells did not show a decrease in LD number and size in comparison with untreated cells (Figure 3C).

## 4. Discussion

To date, there are no studies examining the cellular oxidative metabolic activity of patients with NLSD. The goal of this study was to evaluate if triheptanoin can ameliorate the glycolytic and oxidative metabolism when administered to fibroblasts of patients with NLSD-M. The method used to analyze the oxidative metabolism of affected fibroblasts has been widely used in other studies, mainly for oncological pathologies [7,8,10], but has never been used for the study of NLSD. The results demonstrate that triheptanoin ameliorates the metabolic performances in NLSD-M fibroblasts.

Basal respiration represents a variable percentage of the maximal respiratory capacity and is characteristic of each cell type [12,13]. Our results showed that treatment with triheptanoin could normalize the basal levels of NLSD-M fibroblasts. When necessary to enable energetic adaptation, mitochondrial respiration can suddenly increase to the maximum levels to synthesize more ATP. To achieve this, cells use the spare respiratory capacity (SRC). This process is tightly controlled by the nature and flow of nutrients that can be oxidized in the mitochondrial matrix by the tricarboxylic acid (TCA) cycle [8]. SRC is increased in all NLSD-M patients after treatment with triheptanoin, indicating that acetyl-CoA and Propionyl-CoA, the metabolized products of triheptanoin, provide an anaplerotic effect by replenishing deficient TCA cycle intermediates in NLSD-M. SRC represents a particularly robust functional parameter to evaluate mitochondrial reserve. It characterizes the mitochondrial capacity to meet extra energy requirements and represents a determination of mitochondrial fitness [13]. In NLSD-M, both inter- and intra-familial variability in clinical phenotypes can be observed. As physical endurance training and starvation can improve SRC levels [14], different degrees of clinical severity could partially depend on lifestyle, which is known to involve muscle activity, and dietary regimen [6].

The importance of anaplerotic reactions for the regulation of amino acid, glucose, and FA metabolism is unchallenged. Our results suggest that FA metabolism is interrupted in NLSD-M patients due to impaired anaplerosis of the TCA cycle, as no long-chain FA can be used. Triheptanoin derivatives may function as an alternative source of anaplerotic substrates for the TCA cycle, bypassing the block.

To produce energy, the electrons from substrate oxidation generated a gradient proton across the inner mitochondrial membrane, but oxidative phosphorylation is incompletely ‘coupled’, since protons can ‘leak’ across the inner membrane balancing the gradient, without ATP synthesis. Leaked protons link reactive oxygen molecules to produce H2O and thermic dispersion to regulate physiological processes, such as thermogenesis and ROS reduction [15]. The slight increase in proton leak levels observed in NLSD-M patients after treatment with triheptanoin could be beneficial for patients because it contribute to reducing ROS production in the cells.

The balance between glycolysis and oxidative phosphorylation is believed to be critical for maintaining cellular bioenergetics. We observed that in NLSD-M fibroblasts, triheptanoin also stimulates glycolysis. During starvation, when the rates of lipolysis are highest, a major fraction (up to 30%) of the free fatty acids generated from triglyceride breakdown is re-esterified back to triglyceride in adipose tissue [16,17]. This process requires a source of 3-glycerol phosphate, which is generally supplied by glucose via glycolysis. It is possible that these mechanisms are involved in the stimulation of glycolysis by triheptanoin in NLSD-M.

Data obtained after a short triheptanoin treatment revealed that two different NLSD-M cell lines were able to grow faster than untreated cells. These fibroblasts were collected from two siblings, compound heterozygotes for two different missense mutations (p.L56R and p.I193F) [6]. A functional study performed to clarify the pathogenic effect of these variations showed that both missense mutations caused a partial loss of ATGL function [18]. In this case, supplementation with triheptanoin can provide an additional fat source that can be utilized to increase cellular energy production. The greater amount of energy could balance the partial decrease in TAG hydrolysis and ameliorate cell viability. On the contrary, in NLSD 4 and 5 patients, no beneficial effects on cell viability were observed. Genetic analysis of NLSD 4 subjects displayed an ATGL frameshift mutation (c.695delT), which determines no protein production [6]. Although triheptanoin supplementation can increase energy level, it could not be sufficient to compensate for the total loss of ATGL function. No functional study has been performed to clarify the impact of ATGL mutation on protein function in NLSD 5. It could be hypothesized that, also in this case, the treatment with triheptanoin is not sufficient to balance the effect of gene mutation on lipase activity.

Finally, triheptanoin treatment did not show any beneficial effect on lipid accumulation. As expected, the amount of TGs remained constant after the triheptanoin addition, because it does not exert any lipolytic function on neutral lipids stored inside LDs.

The severe clinical expression of myopathy due to *ATGL* gene mutation is probably caused by various pathogenic mechanisms. We postulated that an altered cellular metabolism can participate in myopathic damage. We hypothesized that a reduction in the availability of mono- and diglycerides, due to the impairment of ATGL lipase function, could remove metabolites from the Krebs cycle and beta-oxidation. Our findings are in line with previous studies, suggesting that triheptanoin in NLSD-M produces acetyl-CoA and propionyl-CoA. Propionyl-CoA can be further converted into succinyl-CoA, an anaplerotic substrate for the tricarboxylic acid cycle, supporting mitochondrial energy production.

The limitations of the study are the small sample size obtained from patients suffering with this ultra-rare disease. The results of this experiment cannot be translated directly into humans but constitute an interesting prerequisite for understanding the pathology and may indicate a therapeutic strategy.

## Figures and Tables

**Figure 1 biomolecules-13-00452-f001:**
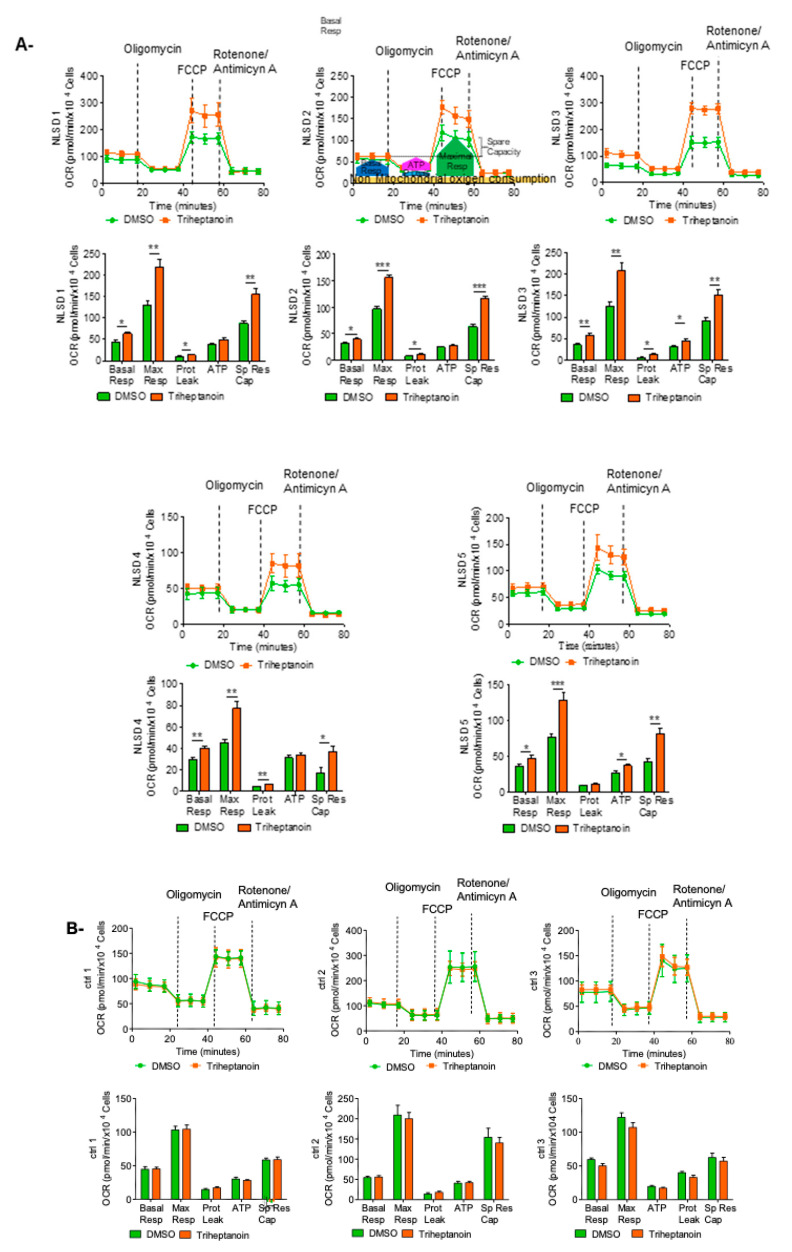
Mitochondrial respiration of fibroblasts treated with triheptanoin. Profile of the mitochondrial activity measured in (**A**) NLSD fibroblasts. The colored frames represent the equivalent areas to calculate basal respiration (BR), proton leak (PL), maximal respiration and non-mytochondrial oxygen consumption (NMO_2_C) values. (**B**) normal fibroblasts. The histograms represent basal respiration, maximum respiration, mitochondrial ATP, Proton Leak and Spare respiratory capacity. Data are presented as mean ± SD. (**C**) comparation of mitochondrial respiration parameters between controls (CTRLs) and NLSD-M samples. (**D**) Comparison of Maximal Respiration and SRC between CTRLs and NLSD-M with ATGL residual activity. Statistical significance was evaluated through Mann–Whitney test. * *p* ≤ 0.05, ** *p* ≤ 0.005, *** *p* ≤ 0.0005.

**Figure 2 biomolecules-13-00452-f002:**
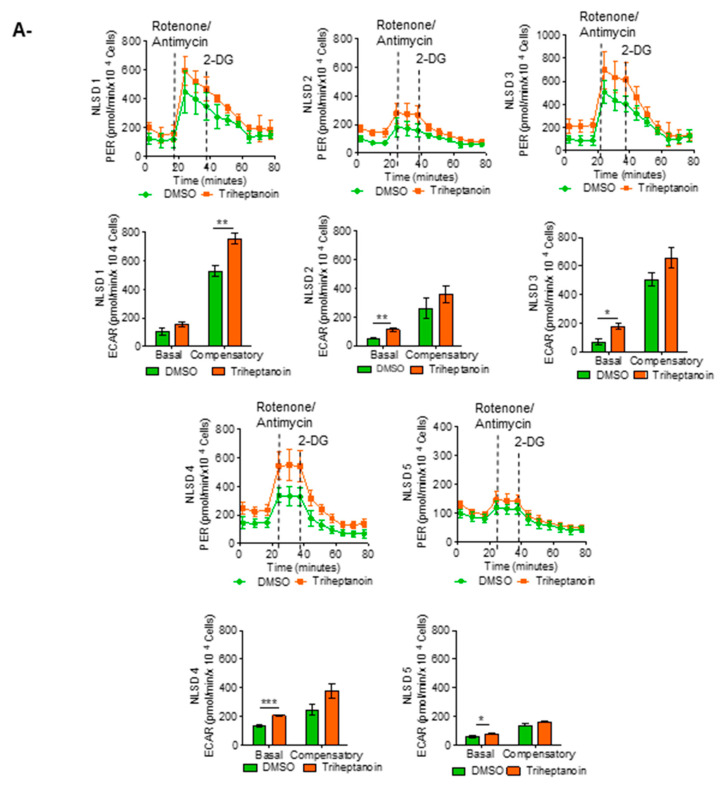
Glycolytic respiration of fibroblasts treated with triheptanoin. Profile of the glycolytic activity measured in (**A**) NLSD fibroblasts and in (**B**) normal fibroblasts. Histograms represent basal glycolysis and compensatory glycolysis. (**C**) comparation of glycolytic respiration parameters between controls (CTRLs) and NLSD-M samples. Data are presented as mean ± SD. Statistical significance was evaluated through Mann–Whitney test. * *p* ≤ 0.05, ** *p* ≤ 0.005, *** *p* ≤ 0.0005. PER (proton efflux rate).

**Figure 3 biomolecules-13-00452-f003:**
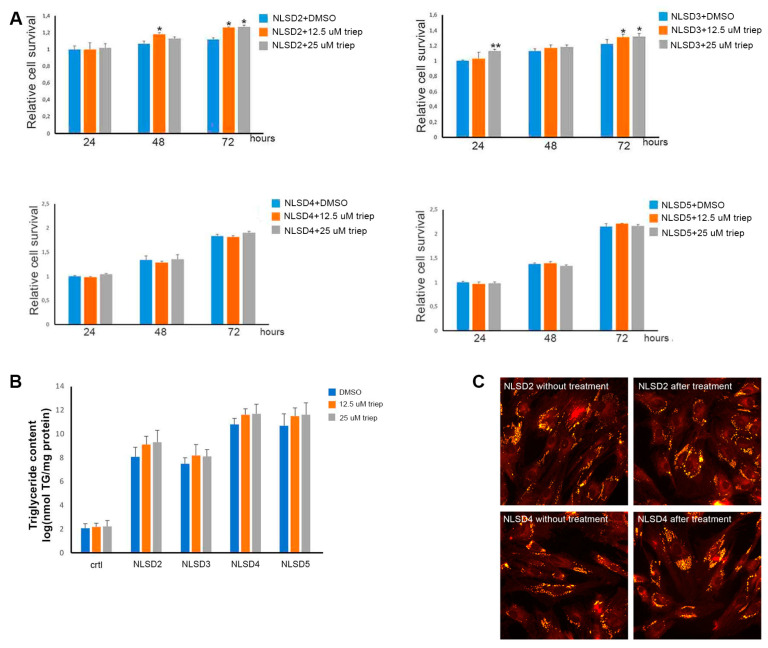
Effect of triheptanoin on cell proliferation and neutral lipid metabolism. (**A**) Relative cell survival measured in NLSDM fibroblasts treated with 12.5 and 25 μM triheptanoin. Histograms represent basal cell viability obtained from 3 independent experiments. Data are presented as mean ± SD. Statistical significance was evaluated through Student’s *t*-test. * *p* ≤ 0.05, ** *p* ≤ 0.001; (**B**) quantification of TAG amount in NLSDM fibroblasts treated with 12.5 and 25 μM triheptanoin. Histograms represent TAG content from 3 independent experiments. Data are presented as mean ± SD; (**C**) NLSDM fibroblasts before treatment and cultured with 25 μM triheptanoin for 9 days and stained with Nile Red. Magnification: 40×.

**Table 1 biomolecules-13-00452-t001:** Clinical and genetic characteristics of NLSD patients.

Patient	Sex	Age	Disease Onset	Mutation	Clinical Severity
NLSD 1	M	68 y	40 y	c.24G > C/c.516C > A	moderate disease
NLSD 2	F	51 y	35 y	c.177T > G/.577A > T	moderate disease
NLSD 3	M	58 y	58 y	c.177T > G/.577A > T	very mild disease
NLSD 4	F	58 y	39 y	c.659delT	severe disease
NLSD 5	M	48 y	40 y	c.45_47del	mild disease

Moderate disease = significant muscle deficit in the 4 limbs, but able to walk; very mild disease = very mild deficit in few muscles; severe disease = severe involvement of 4 limbs and loss of ambulation; mild disease = exclusive involvement of the muscles of the upper limbs. C659del T is a mutation associated with null ATGL production.

**Table 2 biomolecules-13-00452-t002:** Mitochondrial Respiration.

OCR (pmol/min/×10^4^ Cells)	Basal			Maximal Respiration			Proton Leak			ATP			Spare Respiratory Capacity		
	DMSO	Trihep	*p*	DMSO	Trihep	*p*	DMSO	Trihep	*p*	DMSO	Trihep	*p*	DMSO	Trihep	*p*
NLSD 1	44 ± 14	63 ± 10	0.01	130 ± 22	218 ± 44	0.006	9 ± 5	13 ± 5	ns	38 ± 7	50 ± 9	0.02	87 ± 14	155 ± 36	0.0002
NLSD 2	33 ± 4	39 ± 7	0.02	96 ± 16	155 ± 16	0.0001	8 ± 3	12 ± 4	0.01	25 ± 5	28 ± 4	ns	63 ± 14	116 ± 64	0.0001
NLSD 3	35 ± 9	57 ± 15	0.01	125 ± 28	209 ± 43	0.002	5 ± 3	12 ± 5	0.01	30 ± 7	45 ± 12	0.02	90 ± 21	152 ± 29	0.002
NLSD 4	30 ± 5	40 ± 7	0.0003	45 ± 8	77 ± 23	0.007	4 ± 1	6 ± 1	0.001	31 ± 8	34 ± 6	ns	22 ± 18	37 ± 17	0.03
NLSD 5	35 ± 11	47 ± 11	0.03	77 ± 12	129 ± 29	0.0004	9 ± 2	11 ± 5	ns	27 ± 11	37 ± 8	0.03	46 ± 16	81 ± 23	0.001
Normal 12 y	45 ± 10	45 ± 7	ns	103 ± 14	104 ± 18	ns	15 ± 4	17 ± 5	ns	30 ± 7	28 ± 3	ns	59 ± 7	59 ± 11	ns
Normal 53 y	55 ± 8	56 ± 10	ns	209 ± 69	200 ± 41	ns	14 ± 9	18 ± 8	ns	41 ± 13	42 ± 9	ns	159 ± 17	140 ± 36	ns
Normal 73 y	59 ± 8	50 ± 11	ns	122 ± 23	107 ± 23	ns	20 ± 5	17 ± 4	ns	40 ± 7	33 ± 11	ns	62 ± 22	57 ± 19	ns

OCR (oxygen consumption rate). Values represent the mean ± SD. Statistical analysis by Mann–Whitney *t*-test. Trihep (triheptanoin).

**Table 3 biomolecules-13-00452-t003:** Mitochondrial respiration of CTRL vs. NLSD-M fibroblasts.

OCR (pmol/min/×10^4^ Cells)	CTRL		NLSD-M			
	DMSO ^1^	Triheptanoin ^2^	DMSO ^3^	Triheptanoin ^4^	*p* (^1^ vs. ^3^)	*p* (^1^ vs. ^4^)
Basal Oxphos	53 ± 8	50 ± 5	35 ± 5	49 ± 10	0.03	ns
Maximal Resp	145 ± 56	137 ± 54	95 ± 35	158 ± 58	ns	ns
SRC	93 ± 57	85 ± 47	31 ± 29	108 ± 50	ns	ns
ATP	37 ± 6	34 ± 7	30 ± 5	39 ± 9	ns	ns
Proton Leak	16 ± 3	17 ± 7	7 ± 2	11 ± 3	0.03	0.04

OCR (oxygen consumption rate); ^1^ (CTRL, DMSO); ^2^ (CTRL, Triheptanoin); ^3^ (NLSD-M, DMSO); ^4^ (NLSD-M, Triheptanoin). Values represent the mean ± SD. Statistical analysis by Mann–Whitney *t*-test.

**Table 4 biomolecules-13-00452-t004:** Glycolytic respiration.

ECAR (pmol/min/×10^4^ Cells)	Basal Glycolysis			Compensatory Glycolysis		
	DMSO	Triheptanoin	*p*	DMSO	Triheptanoin	*p*
NLSD 1	106 ± 71	154 ± 48	ns	529 ± 96	755 ± 84	0.004
NLSD 2	50 ± 7	112 ± 27	0.004	258 ± 160	359 ± 144	ns
NLSD 3	68 ± 37	175 ± 55	0.02	504 ± 102	655 ± 186	ns
NLSD 4	135 ± 31	206 ± 15	0.0001	245 ± 106	379 ± 146	ns
NLSD 5	60 ± 14	78 ± 13	0.03	139 ± 23	164 ± 21	ns
ctrl 1	148 ± 60	188 ± 108	ns	319 ± 70	349 ± 160	ns
ctrl 2	271 ± 100	277 ± 127	ns	624 ± 129	595 ± 133	ns
ctrl 3	58 ± 11	69 ± 15	ns	157 ± 40	192 ± 38	ns

## Data Availability

The data that support the findings of this study are available from Nelida Inés Noguera, Santa Lucia Foundation, Rome, Italy.

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
