# Peer review of "Effects of Triheptanoin on Mitochondrial Respiration and Glycolysis in Cultured Fibroblasts from Neutral Lipid Storage Disease Type M (NLSD-M) Patients"

_biomolecules, 2023, doi:10.3390/biom13030452_

Round 1

Reviewer 1 Report

This is a biochemical study about an ultrarare recessive neutral lipid storage disease which causes lipid accumulation in all tissues of the body.

Analyses were made from five patients by skin biopsy. Then mitochondrial respiration was measured by single cell Seahorse technology. 

The effect of the medium -chain triglyceride Triheptanoin was shown and proved to effective for respiration and may function as alternative source of anapldrotic substrates for the Krebs cycle. No effect on lipid accumulation was shown.

Here the time courses should be discussed more: cell experiments versus lipid storage over years in patients!

Minor errors: please explain ATGL also in the Abstract (line 29)

In line 32 the Italian "e" should be replaced by "and"

Author Response

REVIEWER 1

We appreciate your review and suggestions. We modified the manuscript in accordance with the requests, hoping to have satisfied the indications.

  1. Please explain ATGL also in the Abstract (line 29)  RESPONSE: we have added the full name of the enzyme, whose function is described later in the text
  2. In line 32 the Italian "e" should be replaced by "and". RESPONSE: it was corrected

Reviewer 2 Report

The article analyze the role of triheptanoin (UX007), a medium chain triglyceride, that, through its anaplerotic effect, could rescue energy production in fibroblast from neutral lipid storage disease type M (NLSD-M) patients.  This rare disease is characterized by skeletal muscle and cardiac damage and large lipid droplets in cytoplasm do to defective ATGL enzyme and consequent reduction of mono (MG) and diglycerides (DG) substrates of Krebs cycle.

Glycolytic and mitochondrial functions, cellular viability and triglyceride content were determined to clarify the possible anaplerotic effect and energy rescue of UX007 and its possible therapeutic effect since there are currently no therapies for this disease while triheptanoin is approved from FDA for patients with long-chain fatty acid oxidation disorders.

To justify the use of triheptanoin in the introduction it should be explained what is the rationale for using this synthetic triglyceride in the absence of the ATGL enzyme. In the case of the cited long-chain fatty acid oxidation disorders what kind of reactions are triggered by treatment.

In table 1 the authors should indicate patient sex.  In addition, given that “causes a severe skeletal and cardiac muscle damage with lipid accumulation” cardiac defect should be referred. It should be better to complete informations with mutation and possibly relasionship of the patients.

Line 89 DMEM medium is high or low glucose?

Line 98 how many wells for cell culture microplates?

From Line 106 the authors illustrate the Oligomycin and FCCP action, not the same for rotenone/antimycin A

Line 108 “This reduction in OCR is linked to cellular ATP production” What does it means functionally?

Authors should discuss if the use of MTT assay to quantify cell proliferation and viability is appropriated in patient cells because the reagent should pass cell and mitochondrial membranes to be reduced by metabolically active cells. Given that “an altered functioning of the aerobic mitochondrial metabolism was also involved in the pathogenesis of the disease” probably MTT is not the best test for these experiments, or, at least, MTT results must be validated by a viability test independent of mitochondrial metabolism.

Line 131 the authors should explain why NLSD-M and control fibroblasts were cultured in Earle’s Basal Medium instead of the previous DMEM and then, at Line 139, in Earle’s MEM culture.

Line 137 “2.5 Immunofluorescence analysis” It should be better to explain  Immunofluorescence analysis of LDs

Fig 1 In my opinion it is not clear how ATP and proton leak are calculated

Line 203 where does the total defect in ATGL enzymatic activity of NLSD4 patient appear?

Fig 2A NLSD1 patient in the upper graph are indicated by opposite colors, the same for ctrl 3 graphs.  The measurement unit of y-axis it is not easily readable and it is not clear, from methods, the difference between y-values in linear graph and corresponding histograms

Lines 258-259 “The results showed that UX007 did not increase lipolysis rate.” The authors should refer only about triglyceride content and not on lipolysis rate

Fig 3C Does Red Nile staining correspond to yellow droplets? What do the red and yellow colors stand for?

SRC should be added in the abbreviation list

Lines 319-320 “2 different NLSD-M cell lines” does it refer to the fibroblasts of two patients? In particular two sibilings?

How “UX007 is useful to increase the glucose and lipid metabolism”?

 In my opinion the paper requires some clarification and addition. The topic is of relevance and the proposed therapeutic approach by repurposing an FDA approved drugs could be of interest.

Author Response

REVIEWER 2

We very appreciate your suggestions to meliorate the manuscript. We modified the manuscript in accordance with the requests, hoping to have satisfied all the indications.

  1. To justify the use of triheptanoin in the introduction it should be explained what the rationale is for using this synthetic triglyceride in the absence of the ATGL enzyme. In the case of the cited long-chain fatty acid oxidation disorders what kind of reactions are triggered by treatment: thank you for making this comment. The following paragraph was added to introduction “The MCT diet plus carnitine supplement in this disorders stimulate citric acid cycle function and ATP production, enhance gluconeogenesis and urea cycle function and maintain intra-mitochondrial homeostasis” and “Triheptanoin is a triglyceride of three odd-chain fatty acid (heptanoate, C7), it is cleavage by intestinal lipases and heptanoate is absorbed by the gastrointestinal tract. Intracellularly, C7 cross the mitochondrial membranes without an active transport system and can enter the β-oxidation cycle. The rational to use triheptanoin in patient with NLSD-M is based on the hypothesis that energy deficiency in fatty acid oxidation may be exacerbated by the depletion of catalytic intermediates of the TCA and would thus benefit from the anaplerotic effect of triheptanoin as well as to its ability to bypass metabolic block induced by ATGL deficiency”. Lines 67 to 69 and 72-79.

  1. In Table1 the authors should indicate patient sex: we added in Table1 this information.

  1. In addition, given that “causes a severe skeletal and cardiac muscle damage with lipid accumulation” cardiac defect should be referred: in Material and Methods we added “None of the patients had severe cardiomyopathy, nor did they wear pacemakers.”
  2. It should be better to complete the informations with mutation and possibly relationship of the patients: in the text we added that two of patients are siblings. The mutations are listed in Table 1 in the original version of manuscript.
  3. Line 89 DMEM medium is high or low glucose? Thank you for the comment, DMEM high glucose was added Line 107.
  4. Line 98 how many wells for cell culture microplates? Thank you for the comment, “four wells for each condition” was added. Lines 116.
  5. From Line 106 the authors illustrate the Oligomycin and FCCP action, not the same for rotenone/antimycin A Thank you for making this comment, “Rotenone, a complex I inhibitor, and antimycin A, a complex III inhibitor, shuts down mitochondrial respiration enabling the calculation of non-mitochondrial respiration” was added. Lines 128-129.
  6. Line 108 “This reduction in OCR is linked to cellular ATP production” What does it means functionally? The functional meaning is based on that, oxygen consumption is directly proportional to mitochondrial respiration and ATP production, then the inhibition of ATP synthase (complex V) by oligomycin produce a decrease in oxygen consumption allowing us to calculate how much ATP the cell produces by mitochondrial respiration.
  7. Authors should discuss if the use of MTT assay to quantify cell proliferation and viability is appropriated in patient cells because the reagent should pass cell and mitochondrial membranes to be reduced by metabolically active cells. Given that “an altered functioning of the aerobic mitochondrial metabolism was also involved in the pathogenesis of the disease” probably MTT is not the best test for these experiments, or, at least, MTT results must be validated by a viability test independent of mitochondrial metabolism. Thank you for this interesting comment. In NLSD-M cells the most important problem for mitochondrial metabolism is the low availability of substrate for energy production due to TAG metabolism defects. Indeed, in these cells there are no dysfunctions of mitochondria proteins or modifications of their structure. The treatment with a compound able to bypass ATGL pathway should provide Krebs cycle reagents and restore mitochondria metabolic activity. Therefore, we assumed that the use of MTT assay was the best choice to highlight the increase in metabolic activity in NLSDM fibroblasts after triheptanoin treatment. In addition, we also evaluated cell number, after some treatments (data not shown), to confirm the results obtained by MTT assay.
  8. Line 131 the authors should explain why NLSD-M and control fibroblasts were cultured in Earle’s Basal Medium instead of the previous DMEM and then, at Line 139, in Earle’s MEM culture. We thank the reviewer for this observation. We made a mistake in reporting “Earle’s Basal Medium” and “Earle’s MEM culture” because fibroblasts were cultured in DMEM. The mistake has been corrected.
  9. Line 137 “2.5 Immunofluorescence analysis” It should be better to explain Immunofluorescence analysis of LDs. Following the reviewer suggestion, we explained in depth Immunofluorescence analysis of LDs: “….and analyzed using the public domain Java image processing program ‘WCIF ImageJ 1.35j’ (developed by W. Rasband, NIH, Bethesda, MD, USA). This software allows us to isolate components with the same wavelength and to measure and quantify different parameters like area, numbers of selected units (LDs, in this case) and pixels per cell; 15 inches2 was chosen as threshold value for LD area to discard fluorescent emissions due to impurity.”
  10. Fig 1 In my opinion it is not clear how ATP and proton leak are calculated Thank you for making this comment, the ATP and proton leak are calculated by a software analysis. The decrease in oxygen consumption rate upon injection of the ATP synthase inhibitor oligomycin represents the portion of basal respiration that was used to drive ATP production. Proton Leak is the remaining basal respiration not coupled to ATP production. A graphical representation to clarify this point was added to Figure 1A.
  11. Line 203 where does the total defect in ATGL enzymatic activity of NLSD4 patient appear? Thank you for making this comment, “C659del T is a mutation associated to null ATGL production” was added to the legend of Figure 1.
  12. Fig 2A NLSD1 patient in the upper graph are indicated by opposite colors, the same for ctrl 3 graphs. The measurement unit of y-axis it is not easily readable, and it is not clear, from methods, the difference between y-values in linear graph and corresponding histograms. Thank you for marking this error, the graph in Fig 2A and ctrl 3 were corrected. The Y-axis was changed with PER (proton exclusion rate) in pmol/min to homogenize with the corresponding histograms.
  13. Lines 258-259 “The results showed that UX007 did not increase lipolysis rate.” The authors should refer only about triglyceride content and not on lipolysis rate. Thank you for this comment. We quantified cellular TAG content, and we used these measurements as an indirect evaluation of lipolysis rate. We understand that our sentence can be misleading. We modified the sentence: “The results showed that triheptanoin did not decrease neutral lipid abnormal storage”.
  14. Fig 3C Does Red Nile staining correspond to yellow droplets? What do the red and yellow colors stand for? Thank you for this advice. Yellow droplets correspond to Nile Red staining, while red staining was due to an overexposed image and, sometimes, to the formation of Nile Red aggregates on the coverslip surface. We changed some photo parameters to improve image quality.

  1. SRC should be added in the abbreviation list. In the policy of journal there isn’t an “Abbreviation list”, the explanation of the abbreviations is in the text.
  2. Lines 319-320 “2 different NLSD-M cell lines” does it refer to the fibroblasts of two patients? In particular two siblings? Yes, the fibroblasts were obtained from biopsies of two siblings, as clarified in the text, line…, “These fibroblasts were collected from 2 siblings…”.
  3. How “UX007 is useful to increase the glucose and lipid metabolism”? As explained above, triheptanoin is an alternative substrate able to replenish the intermediates of Krebs cycle. Indeed, triheptanoin can provide both acetyl-CoA and propionyl-CoA, two molecules essential for mitochondrial ATP production. Initial studies demonstrated anaplerotic effect of this compound in long-chain fat oxidation disorders. Further researchers also shown that in case of glucose metabolism disorders, such as glucose transporter 1 deficiency syndrome (GLUT1-DS) and pyruvate carboxylase deficiency, defects of glucose transporter could be compensated by the supplying of Krebs cycle intermediates from heptanoate and/or C5-ketone bodies.

Reviewer 3 Report

The authors have generated novel and interesting data. They are the first to culture cells from 5 affected individuals with a Neural Lipid Storage Disease and investigate the effect of an FDA approved drug on glycolytic and mitochondrial function. 

The manuscript itself is very well written. The science is sound. The methodology reported was concise and reproducible. Also, the results were clearly articulated and supported by the figures/tables.  Overall, I enjoyed reading this paper. 

Minor corrections:

Line 27 - add autosomal recessive.

Lines 38, 288, 291 - change reduce font size of "triheptanoin"

Line 82 - I"m assuming that the PNPLA2 mutation status of the control fibroblasts was confirmed by sequencing 

Graphs - On the small size and need to be a little larger (the resolution was not great when I used the magnify button).

Line 274 - remove "have" and replace with "were"  

Line 277 - remove comma (12,5) and add a dot (12.5). This formatting is also present in figure 3.

Author Response

REVIEWER 3

We are very grateful for the review. We modified the manuscript in accordance with the requests, hoping to have satisfied all the indications.

  1. Line 27 - add autosomal recessive: we added “autosomal”.

  1. Lines 38, 288, 291 - change reduce font size of "triheptanoin": this it was corrected

  1. Line 82 - I’m assuming that the PNPLA2 mutation status of the control fibroblasts was confirmed by sequencing: we have checked for the ATGL gene mutations in the control fibroblasts. Furthermore, we have excluded the existence of a myopathy in the controls with clinical history, serum CK assay and the neurological examination. We know, from the literature and from our experience (reference 6), that to date all patients with NLSD-M have hyperCKemia. We added in the text that the controls were evaluated for the possible existence of a myopathy.

  1. Graphs - On the small size and need to be a little larger (the resolution was not great when I used the magnify button. The graph was enlarged.

  1. Line 274 - remove "have" and replace with "were": this it was corrected

  1. Line 277 - remove comma (12,5) and add a dot (12.5). This formatting is also present in figure 3. Thanks for this observation, this error was corrected in the text.